# Phenotypic Presentations of Cystic Fibrosis in Children of African Descent

**DOI:** 10.3390/genes12030458

**Published:** 2021-03-23

**Authors:** Sophie Mayer Lacrosniere, Michele Gerardin, Laurence Le Clainche-Viala, Veronique Houdouin

**Affiliations:** AP-HP, Pediatric Pulmonology and CF Centre, University Hospital Robert Debre, University of Paris, CEDEX 19, 75935 Paris, France; michele.gerardin@aphp.fr (M.G.); laurence.leclaincheviala@aphp.fr (L.L.C.-V.); veronique.houdouin@aphp.fr (V.H.)

**Keywords:** cystic fibrosis, children, Africa

## Abstract

The Robert Debre Pediatric Cystic Fibrosis (CF) centre, located in the North East of Paris, a multicultural area, is in charge of a cohort of around a hundred and sixty children diagnosed with CF. Between 2000 and 2019, the proportion of children of African descent in this centre increased from 2% to 10%. We report the clinical features of 17 children of African descent diagnosed with CF: 4 (23%) were diagnosed after a meconium ileus, 14 (83%) had exocrine pancreatic insufficiency, and 7 (41%) had early *Pseudomonas aeruginosa* infection before the age of two. Even though the majority of patients were diagnosed through NBS, the twenty-nine-mutation testing kit proved less effective in non-Caucasian populations, with a false negative rate of 25% in this series. CF is definitely not solely a Caucasian disease and the literature reveals similar phenotypes in Caucasian and African people provided that they present the same CFTR mutations. Clinicians have to keep in mind that the diagnosis of CF in patients of African descent must be evoked in the case of symptoms and a sweat test must be performed, despite a negative result for NBS.

## 1. Introduction

Cystic fibrosis is a genetic disease with a multisystemic expression affecting the lungs, the pancreas, the intestinal tract, and the ENT sphere. An early diagnosis is important to prevent or delay complications and pulmonary decline.

The Robert Debre Pediatric CF centre, located in the North East of Paris, a multicultural area, is in charge of a cohort of children of African descent with CF. The fact that CF has been extensively studied in individuals of Caucasian origin led to the misconception that this disease specifically affects the Caucasian population. A recent review of the literature [1,2] found reports from only 12 of the 54 African states on the molecular analysis of the mutations present in suspected CF patients, resulting in the identification of 79 mutations and among them 39 proven to cause CF and 21 unique to Africa.

The Newborn Screening (NBS) programme for CF was introduced in 2002 in France and relies on an immunoreactive trypsinogen (IRT) and a 29-mutation genetic screening [3]. NBS marked a turn in the care of CF, especially for children of African descent: since 2002 we observed an increase in the proportion of children from African descent in our cohort from 1% in 2000 before NBS to 10% in 2019.

The objective is to describe the phenotypic presentations of cystic fibrosis in children with at least one parent from Western or Central Africa (WCA) and to discuss the diagnostic challenges in this population [4,5].

## 2. Materials and Methods

We conducted a retrospective cohort study between 2000 and 2019 at Robert Debre Pediatric CF centre. We conducted two cross analyses in 2000 before NBS implementation and in 2019 in order to illustrate the emergence of a population of African descent after the generalisation of NBS. Every child with a diagnosis of CF with at least one parent from Western or Central Africa (WCA) was included. We chose not include patients from the French overseas departments and from Guiana and Haiti in order to describe a less heterogeneous population. Data concerning the nature of the CFTR mutations, the circumstances of diagnosis, the presence of an exocrine pancreatic insufficiency, and age at first isolation of *Pseudomonas aeruginosa* were collected.

## 3. Results

Between 2000 and 2019, the number of children with CF attending our centre remained stable, with around 160 patients from an ethnically diverse population.

In 2000, 168 children attended the CF clinic. Three cases from two families had at least one parent from WCA (3/168 2%). One case was diagnosed after a meconium ileus and subsequently in the sibling. The third child was diagnosed at eighteen months with a delay in diagnosis of twelve months after the first clinical manifestations (failure to thrive, respiratory symptoms).

In 2002, the NBS was generalized in France. The program is organized in two steps: a screening based on the level of TIR at three days of life and the use of a genetic panel with the 29 most frequent mutations in France.

In 2019, the cohort was made of 165 children including 17 cases (15 families) with at least one parent from WCA (17/165, 10%). Among those 17 cases, one was already included in the cohort in 2000, diagnosed before NBS, and the subsequent 16 patients born after 2002 benefited from NBS. Among the 16 patients diagnosed after 2002, twelve (75%) were diagnosed using NBS. The NBS false negative rate was 25%. Among the four NBS false negatives, two were associated with a meconium ileus. The overall proportion of meconium ileus was 19% (n = 3/16).

The circumstances of diagnosis, the country of origin of the parents, the CFTR mutations, the symptoms in the first year of life, presence of a pancreatic insufficiency, and the age of first *Pseudomonas aeruginosa* isolation are detailed in Table 1 for all the patients with at least one parent from WCA diagnosed or followed since 2000 in our centre.

Seven out of 16 patients (44%) had no mutation included in the NBS 29 mutations kit.

## 4. Discussion

The proportion of children with CF with one or two parents from WCA increased from 2% in 2000 to 10% in 2019. This increase may be explained by underdiagnosis in this population before NBS. Before 2000, children of African descent were all diagnosed based on the clinical manifestations of the disease. One was diagnosed after a meconium ileus, the sibling was diagnosed at birth due to the family history, and the third was diagnosed at the age of 18 months. The delay in diagnosis exceeded twelve months after the onset of the symptoms, despite typical clinical manifestations—the diagnosis was not suggested because of his ethnic origin due to a common misconception that CF mainly affects Caucasian people [2,5]. Presumably, children with CF of African descent were largely under-diagnosed before the generalization of NBS. It is nevertheless difficult to precisely establish to what extent the variations in the flow of migration impacted our observations.

The NBS generated a marked increase in the CF diagnosis in African children, as illustrated by the analysis of the 2000 and 2019 cohort. However, the elevated rate of NBS false negatives (25%) raises our attention, contrasting with the proportion in the general population expected at below 5% [6]. Out of sixteen patients, seven had no mutations among the twenty-nine assessed. Two false negatives were associated with meconium ileus in the link with lower levels of immunoreactive trypsin. For these patients, the genotype was identified a second time by an extensive screening of CFTR gene. Seven out sixteen patients (44%) had no mutation included in the NBS 29 mutations kit—in comparison, among all patients diagnosed in France in the same period, 3.9% had no mutations identified by the NBS kit [7]. This confirms the poorer sensitivity of the NBS kit, which was tailored for the Caucasian population, in the African population, as previously described in the Afro-American population [4,8,9].

Regarding the phenotype, 83% (n = 14) of our patients had exocrine pancreatic insufficiency, similar to non-African CF patients.

The elevated rate of meconium ileus is to be noticed (17%, n = 3), even though on a small sample; nationwide, in 2018, the rate of meconium ileus was 8% [10]. The data from the literature concerning the incidence of meconium Ileus are heterogeneous. For example, in the American population, the rate of meconium ileus is about 20% [11].

In our cohort, 41% of the patients presented an early *Pseudomonas aeruginosa* isolation in the first two years of life. Unfortunately, we cannot present an evaluation of the pulmonary function because of the limited size of our cohort and the heterogenicity of the ages of the children.

A literature review finds two case–control studies comparing the phenotypes of Caucasian and African patients with the same genotype F508del homozygote [12,13]. An American study based on the US CF Foundation National Patient Registry [12] found that patients of African descent with CF were younger at diagnosis and had poorer nutritional status and pulmonary function than white patients with CF, fewer had meconium ileus, but more have distal intestinal obstruction syndrome. To control for genotype, each patient of African descent’s F508del homozygote was compared with four age- and sex-matched Caucasian F508del homozygotes, with only the difference in nutritional status remaining. Both of these studies [12,13] concluded that for the same mutations, phenotypic presentation is comparable between African patients and non-African controls. This suggests that the phenotypic differences between patients of African and Caucasian descent may essentially be due to the genetic heterogeneity of CFTR mutations.

## 5. Conclusions

CF is definitely not only a Caucasian disease, and this must be emphasized among clinicians.

The literature shows similar phenotypes in patients of African and Caucasian descent, as long as they have the same CFTR mutations and the same access to healthcare.

Our study shows an increase in the diagnosed cases of CF in the African population since the onset of the NBS program in 2002 in France. False negative results of NBS are more frequent than in the Caucasian population. The panel of 29 mutations detected in the French NBS program is less efficient in this population. The diagnosis of CF in patients of African descent must be considered in the case of typical symptoms despite a negative NBS result.

In the future, the question of the eligibility for CFTR modulators will be discussed because of the presence of rare mutations being more frequent in this population.

## Figures and Tables

**Table 1 genes-12-00458-t001:** Characteristics of children with at least one parent from WCA followed since 2000.

Year of Birth	Sex	Age at Diagnosis	NBS Status	Sweat Test	Symptoms in the First Year of Life	Exocrine Pancreatic Insufficiency	Age at First Isolation of *Pseudomonas aruginosa*	CFTR Mutation 1 (Legacy Name)	CFTR Mutation 2 (Legacy Name)	Father/Mother’s Native Country
2000	F	Neonatal	NC	Positive		yes	9 y	*F508del	S549N	Congo/Italie
2005	F	Neonatal	TP (USA)	Positive		yes	10 y	3745delC	4374 + 1G > A	Mali/Mali
2005	M	6 months	FN	Positive	respiratory symptoms, dehydratation	yes	10 y	R1066H	399insT	Congo/Congo
2007	M	Neonatal	TP (SPAIN)	Positive		yes	3 y	*3120 + 1G > A	*3120 + 1G > A	Congo/Congo
2009	M	Neonatal	TP	Positive	respiratory symptoms	no	no	*3120 + 1 > G	NC	Congo/Congo
2009	M	Neonatal	TP	Positive	failure to thrive	yes	3 y	*F508del	L1234V	Republic of Sierra Leone/France
2010	F	Neonatal	TP	Positive		yes	no	*3120 + 1 > G	*3120 + 1 > G	Congo/Brasil
2010	M	Neonatal	TP	Intermediate		no	no	A96E	A96E	Congo/Central African Republic
2011	M	Neonatal	TP	Intermediate		yes	no	*F508del	273 + 4 A > G	Cameroon/France
2012	M	3 months	FN	Positive	failure to thrive, hypoalbuminemia	yes	0.5 y	CFTRdele 19–21	CFTRdele 19–21	Mali/Mali
2013	F	Neonatal	TP	Positive	meconium ileus	yes	1 y	*R553X	*R553X	Mali/Mali
2014	F	Neonatal	FN	Positive	meconium ileus requiring surgery	yes	1 y	C233 dup	del17a-18	Cote d’Ivoire/France
2014	F	Neonatal	TP	Positive	failure to thrive	yes	0.1 y	CFTRdele 19–21	CFTRdele 19–21	Mali/Mali
2015	M	Neonatal	TP	Positive	failure to thrive	yes	1 y	*R553X	*R553X	Mali/Mali
2015	F	Neonatal	TP	Positive		yes	1 y	*711 + 1G > A	Q552P	Senegal/France
2016	F	Neonatal	FN	Positive	meconium ileus requiring surgery	yes	0.5 y	c4230C > A	c4230C > A	Senegal/Senegal
2018	F	Neonatal	TP	Positive		no	no	*3120 + 1 G > A	F311del	Antilla/Congo

## Data Availability

Data will be available up to two years after publication.

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
