# Peer review of "Phenotypic Presentations of Cystic Fibrosis in Children of African Descent"

_genes, 2021, doi:10.3390/genes12030458_

Round 1

Reviewer 1 Report

The manuscript by Mayer Lacrosniere and colleagues describes the phenotypes of children with CF who are of African descent and draws some comparisons with caucasian populations. The manuscript is simply presented and will be of interest to clinical readers working int he field.

The abstract notes an increase in incidence of 2 to 10% during the course of the study; However, it is acknowledged in the discussion that this may be influenced by the introduction of NBS. It would be interesting to know if a higher rate of diagnosis was present in older children around 2000 when the study began i.e. if children were being diagnosed based on the clinical manifestation of the disease at an older age, rather than as infant. Presumably, children with CF would not go undetected. I think this is an important point to determine if the incidence has truly increased or whether this is simply done to advances in NBS.

Additionally, please expand on the NBS kit and how it is better suited to caucasian populations rather than those of African descent.

A few minor typos: American requires a capital A (Line 93). Sentence on lines 94-95: 'patients' appears twice.

Author Response

Dear Reviewer,

Thank you for you corrections and comments:

The abstract notes an increase in incidence of 2 to 10% during the course of the study; However, it is acknowledged in the discussion that this may be influenced by the introduction of NBS.

It would be interesting to know if a higher rate of diagnosis was present in older children around 2000 when the study began i.e. if children were being diagnosed based on the clinical manifestation of the disease at an older age, rather than as infant 

S Mayer Lacrosniere> Before 2002 CF was only diagnosed in symptomatic childrens or their siblings, out of three patients in the 2000 cohort two were diagnosed in the first months with no delay in diagnosis but one was diagnosed at 18 months with typical symptoms starting more than twelve months before the diagnosis. We corrected the manuscript to explicit these circumstances of diagnoses in the results

Presumably, children with CF would not go undetected. I think this is an important point to determine if the incidence has truly increased or whether this is simply done to advances in NBS: 

S M>the incidence of diagnoses of CF in patients of African descent has increased after the onset of NBS in our centre, we tried to illustrate this by two transversal analyses in 2000 and 2019, the patients diagnosed had typical forms of cystic fibrosis, we believe that a number of children could have gone undetected before 

Additionally, please expand on the NBS kit and how it is better suited to caucasian populations rather than those of African descent

S M>the kit is designed to find the 29 most frequent CFTR mutations, it was designed for a population with a majority of Caucasian people, the people of African descent are a minority and mutations specific of this population are not included in this panel, the same results were found for minorities in the US with reduced specificity of the kit in these minorities. We tried to clarify this point in the discussion

A few minor typos: American requires a capital A (Line 93). Sentence on lines 94-95: 'patients' appears twice.

SM >corrected, thank you, we decided to apply for English Editing (to be completed within 2 days)

We also tried to clarify the conclusion

Please see the attachment:  version of the manuscript integrating the corrections

Reviewer 2 Report

This is a review of the review manuscript entitled, "Phenotypic presentations of cystic fibrosis in children of African descent" by Lacrosniere et al. This study/review highlights the phenotypic features of CF and analyzes how it manifests in the population of cases of African descent. It covers an interesting subject that needs to be reported. The manuscript does need some assistance in English proof-reading to make it flow better and better highlight the results. I would also suggest that in the introduction a paragraph be devoted to a brief overview of major target organs of CF such as lung disease (PMID: 25607428) and how early diagnosis is important to initiate treatments and prevent acceleration of disease. This will help give the reader context.

Line 80 – "equally", are the authors saying their data showed equal incidence of disease? Or perhaps they are saying the African population is "also" affected. Clarify.

Line 84 – "elevated" It is reported in the literature that meconium ileus is about 15-20% in humans (PMID: 25591863).

Line 90 – "wether" should be "whether"

Line 98 – there were 47 cases African descent with F508del homozygote? This number exceeds the cases reported in the paper – clarify.

Line 106 – were any genotypic definitions made for these cases?

Author Response

Dear Reviewer,

Thank you for your corrections and comment : 

The manuscript does need some assistance in English proof-reading to make it flow better and better highlight the results.

>In response we just applied for English Editing, which should be completed within two days

I would also suggest that in the introduction a paragraph be devoted to a brief overview of major target organs of CF such as lung disease (PMID: 25607428) and how early diagnosis is important to initiate treatments and prevent acceleration of disease. This will help give the reader context.

>thank you, we integrated these elements of context at the beginning of the introduction

Line 80 – "equally", are the authors saying their data showed equal incidence of disease? Or perhaps they are saying the African population is "also" affected. Clarify.

> We meant "also" and modified the sentence

Line 84 – "elevated" It is reported in the literature that meconium ileus is about 15-20% in humans (PMID: 25591863).

>In the USA CF Register reported these figures and we provided a recent reference to precise "elevated", but the French CF Register reports 8% of meconium Ileus

Line 90 – "wether" should be "whether"

>Thank you, following your advice we applied for English Editing

Line 98 – there were 47 cases African descent with F508del homozygote? This number exceeds the cases reported in the paper – clarify.

>We were detailing the results of Hamosh et al, we rewrote this that was unclear

Line 106 – were any genotypic definitions made for these cases?

>we then precised that the genotype was identified in a second time by extensive screening of the CFTR gene

Please find attached the version of the manuscript integrating the corrections, we are waiting for the English Editing that should be available within two days
